# Impact on the Clinical Evolution of Patients with COVID-19 Pneumonia and the Participation of the *NFE2L2/KEAP1* Polymorphisms in Regulating SARS-CoV-2 Infection

**DOI:** 10.3390/ijms24010415

**Published:** 2022-12-27

**Authors:** María Elena Soto, Giovanny Fuentevilla-Álvarez, Adrián Palacios-Chavarría, Rafael Ricardo Valdez Vázquez, Héctor Herrera-Bello, Lidia Moreno-Castañeda, Yazmín Estela Torres-Paz, Nadia Janet González-Moyotl, Idalia Pérez-Torres, Alfredo Aisa-Alvarez, Linaloe Manzano-Pech, Israel Pérez-Torres, Claudia Huesca-Gómez, Ricardo Gamboa

**Affiliations:** 1Department of Immunology, Instituto Nacional de Cardiología “Ignacio Chávez”, Juan Badiano No. 1. Col. Sección XVI., México City 14380, Mexico; 2Cardiovascular Line in American British Cowdray (ABC) Medical Center, I.A.P. ABC I.A.P. ABC Sur 136 No. 116 Col. Las Américas, México City 01120, Mexico; 3Department of Physiology, Instituto Nacional de Cardiología “Ignacio Chávez”, Juan Badiano No. 1. Col. Sección XVI., México City 14380, Mexico; 4Department of Biochemistry, Escuela Nacional de Ciencias Biológicas, Instituto Politécnico Nacional (IPN), Manuel Carpio y Plutarco Elias Calles, Col. Miguel Hidalgo, México City 11350, Mexico; 5Critical Care Unit of the Temporal COVID-19 Unit, Citibanamex Center Av. del Conscripto 311, Lomas de Sotelo, Hipódromo de las Américas, Miguel Hidalgo, México City 11200, Mexico; 6Department of Genetic, Hospital Infantil de México “Federico Gómez”, Doctor Márquez 162, Col. Doctores, México City 06720, Mexico; 7Critical Care in American British Cowdray (ABC) Medical Center, I.A.P. ABC I.A.P. ABC Sur 136 No. 116 Col. Las Américas, México City 01120, Mexico; 8Department of Cardiovascular Biomedicine, Instituto Nacional de Cardiología “Ignacio Chávez”, Juan Badiano No. 1. Col. Sección XVI., México City 14380, Mexico

**Keywords:** SARS-CoV-2, COVID-19, pathogenesis, inflammation, *NFE2L2*, *KEAP1*, single nucleotide polymorphism

## Abstract

In patients with severe pneumonia due to COVID-19, the deregulation of oxidative stress is present. Nuclear erythroid factor 2 (NRF2) is regulated by KEAP1, and NRF2 regulates the expression of genes such as *NFE2L2-KEAP1*, which are involved in cellular defense against oxidative stress. In this study, we analyzed the participation of the polymorphisms of *NFE2L2* and *KEAP1* genes in the mechanisms of damage in lung disease patients with SARS-CoV-2 infection. Patients with COVID-19 and a control group were included. Organ dysfunction was evaluated using SOFA. SARS-CoV-2 infection was confirmed and classified as moderate or severe by ventilatory status and by the Berlin criteria for acute respiratory distress syndrome. SNPs in the gene locus for *NFE2L2*, rs2364723C>G, and *KEAP1*, rs9676881A>G, and rs34197572C>T were determined by qPCR. We analyzed 110 individuals with SARS-CoV-2 infection: 51 with severe evolution and 59 with moderate evolution. We also analyzed 111 controls. Significant differences were found for rs2364723 allele G in severe cases vs. controls (*p* = 0.02); for the rs9676881 allele G in moderate cases vs. controls (*p* = 0.04); for the rs34197572 allele T in severe cases vs. controls (*p* = 0.001); and in severe vs. moderate cases (*p* = 0.004). Our results showed that *NFE2L2* rs2364723C>G allele G had a protective effect against severe COVID-19, while *KEAP1* rs9676881A>G allele G and rs34197572C>T minor allele T were associated with more aggressive stages of COVID-19.

## 1. Introduction

The coronavirus SARS-CoV-2 belongs to the genus β-*coronavirus,* which is responsible for the current acute respiratory syndrome, COVID-19. It is known that, in each wave of SARS-CoV-2, the infection presents a different viral variant, and it has been suggested that patients with the Delta variant experience more severe symptoms and are more likely to die from COVID-19 [1]. Regardless of the various damage mechanisms involved, the genetic influence of hosts upon the virus needs to be investigated to determine the susceptibility of patients and their likely severity of disease. Patients infected with SARS-CoV-2 may experience an asymptomatic course, or they may present with mild or very complex symptoms. Mortality rates may vary between different geographical areas and populations [2]. In addition, infected patients may have a history of chronic metabolic diseases such as hypertension, cardiovascular disorders (CVDs), obesity, diabetes mellitus (DM), and hypercholesterolemia. The presence of these comorbidities can aggravate symptoms in the pulmonary system [3]. Viral infections cause damage to multiple cellular mechanisms; there is evidence of deregulation of oxidative stress in addition to the inflammatory processes involved [4,5]. Some studies have found an association between oxidative stress and the severity of COVID-19 symptoms, indicating that the deregulation of the oxidant system amplifies various damage mechanisms that interact with cytokine storms, coagulopathy, and cell hypoxia [6,7].

It is unclear how SARS-CoV-2 infection causes oxidative stress. Cells have different systems to defend themselves from damage and inflammation in response to aberrant oxidative stress. The nuclear factor erythroid 2-related factor 2 (known as NRF2 or NFE2) comprises a pivotal system of cytoprotective responses to oxidative stresses, in which NFE2L2 is a master transcription factor that controls antioxidant enzymes [8]. Additionally, NFE2L2 is negatively regulated by the cysteine-rich protein KEAP1 via proteasomal degradation mediated by the CUL3 (cullin 3)-E3 ubiquitin ligase RBX1 (ring-box 1) complex [9,10]. This process can result in the oxidation of cysteine residues and other proteins involved in viral infection.

However, the mechanisms by which NRF2 acts against SARS-CoV-2 infection and how this virus induces the antioxidant response are not fully understood. The transcription factor NRF2 regulates the expression of different genes involved in cellular defense against oxidative stress, which occurs in various neurodegenerative, metabolic, cardiovascular, and viral diseases. The main route of regulation of NRF2 is through interactions with the KEAP1 protein [11]. KEAP1 must be inhibited, and free NRF2 activates the expression of antioxidant genes and the antioxidant response (ARE) [12,13,14]. Significant modifications in NFE2L2-KEAP1 pathways have been observed in patients with severe infectious progression, various conditions of acute lung injury (ALI), acute respiratory distress syndrome (ARDS), respiratory failure, heart failure, sepsis, and sudden cardiac arrest [15,16]. Genetic variation in *NFE2L2* and genes involved in ALI may affect the induction of the pathway driven by NRF2 and KEAP1, and this would explain the susceptibility to different clinical stages of SARS-CoV-2.

Furthermore, with the purpose to evaluate organ dysfunction, the method of sequential organ failure assessment (SOFA) scoring was developed to provide a means of quantitatively and objectively describing the degree of organ failure over time in individual patients with sepsis. Therefore, the SOFA score was calculated on the day of admission and on all subsequent days of patient treatment.

To date, there have been few studies of the relationship between these genes and COVID-19, and none at all which have analyzed the polymorphisms of these genes involved in oxidative stress. In this study, therefore, we sought to describe the participation of the single nucleotide polymorphisms (SNPs) of the NFE2L2 (rs2364723) localized in the intronic region and KEP1 (rs967688 and rs34197572) genes localized in the 3’UTR region in lung disease in patients with SARS-CoV-2 infection.

## 2. Results

### 2.1. Clinical Characteristics

The present study involved 221 patients. A total of 110 were positive for SARS-CoV-2 infection, and 111 had not suffered from COVID-19 up to the time of sampling (controls). In the case of patients with COVID-19 infection, the population was stratified into two groups: severe infection (51 patients) and moderate infection (59 patients) (Table 1). The mean age was 61 ± 12 for patients with severe COVID-19 and 54 ± 12 years for patients with moderate COVID-19. The mean age of controls was 37 ± 14 years. We found a statistically significant difference when comparing the body mass index (BMI) of patients in the infection and control groups. The BMIs of infected patients were more likely to be higher (*p* = 0.018). When comparing the incidence of obesity, we found a higher frequency in severe (41.2%) and moderate (40.7%) patients compared with control (34.5%) patients. Healthy patients did not present with diabetes, hypertension, or dyslipidemia; however, when analyzing only those patients with severe or moderate COVID-19 infection, severe cases presented a higher frequency of diabetes (47.1% vs. 30.5%) and hypertension (45.1% vs. 35.6%).

### 2.2. Genotypic and Allelic Frequencies of NFE2L2 (rs2364723), KEAP1 (rs9676881), and KEAP1 rs34197572

Table 2 shows the genotypes and allelic frequencies for the three SNPs studied. In *NFE2L2* (rs2364723), the GG genotype and G allele had a lower frequency in all the patient groups. When we compared the groups, we found a significant difference in the GG genotype in the group with severe COVID-19 (4%) compared with the moderate COVID-19 group (24%) (OR 0.13 CI 95% (0.02–0.60), *p* = 0.001); between the severe COVID-19 and control groups (22%) (OR 0.14 CI 95% (0.03–0.65), *p* = 0.001); and for G allele frequency (OR 0.54 CI 95% (0.32–0.89), *p* = 0.02), suggesting that the minor allele could be protective against damage.

In the case of the *KEAP1* polymorphism (rs9676881), the results revealed significant statistical differences in the GG genotype for the severe vs. moderate groups (OR 0.41 CI 95% (0.18–0.92), *p* = 0.04) and for the moderate vs. control groups (OR 2.27 CI 95% (1.11–4.62), *p* = 0.03). In the same way, when comparing the allele frequency, we found a lower frequency of the G minor allele in patients with moderate COVID-19, compared with healthy subjects (OR 1.91 CI 95% (1.03–3.53), *p* = 0.04).

At the same time, the *KEAP1* polymorphism (rs34197572) presented a lower frequency of the TT genotype in all the studied groups compared to CC. When we analyzed the genotype, we found a significant difference in CC genotype frequency in severe COVID-19 patients; specifically, an underrepresentation (80%) compared with the moderate group (97%) (OR 0.07, CI 95% (0.00–0.57), *p* = 0.002) and with the control group (95%) (OR 0.19 CI 95% (0.06–0.60), *p* = 0.003). An opposite tendency was found for the TT genotype, that is, a significant difference between the severe group and the moderate group (*p* = 0.07) and also the control group (*p* = 0.04). With respect to allele frequency, similar differences were found between the severe group and both the moderate and control groups (OR 21 CI 95% (2.83–167), *p* = 0.004) and (OR 4.97 CI 95% (2.05–12.05), *p* = 0.001), respectively).

### 2.3. Overdominant, Dominant, and Recessive Models of NFE2L2 and KEAP1 SNP in COVID-19 Patients

Finally, for the purpose of determining the genotype effect, an analysis was carried out using three different genetic models: overdominant, dominant, and recessive. We found that the SNP *NFE2L2* (rs2364723) had a statistically significant difference in the recessive model when we compared patients with severe COVID-19 infection and controls (OR 6.75 IC 95% (1.53- 29.82), *p* = 0.008). When reviewing the results of the SNP *KEAP1* (rs9676881), we found significant differences for moderate infection vs. control groups in both the overdominant and recessive models ((OR 2.47 IC 95% (1.18–5.19), *p* = 0.02) and (OR 0.43 IC 95% (0.21–0.89), *p* = 0.03), respectively). Statistically significant differences were also found in the SNP *KEAP1* (rs34197572) when comparing the severe COVID-19 group against the controls in the dominant and recessive models ((OR 0.19 IC 95% (0.06–0.60), *p* = 0.005) and (OR 0.20 IC 95% (0.04–0.86), *p* = 0.04), respectively) (Table 3). In addition, an analysis of correlation between genotypes and the clinical variants was performed; however, no significant differences were found, and the data are not shown here.

## 3. Discussion

SARS-CoV-2 infection is known to result in symptoms of wide-ranging severity. Some infected individuals are wholly asymptomatic; others experience severe symptoms that require assisted respiration. Serious symptoms also include dysregulation of cytokines, which can cause severe pneumonia, sometimes with fatal outcomes. Since the beginning of the pandemic, it has been reported that patients with comorbidities such as diabetes, hypertension, and dyslipidemia are more susceptible to infection and more likely to experience severe symptoms [17]. In addition to major inflammatory participation, oxidative stress has been confirmed in patients with pneumonia, and the response of patients to antioxidant therapy has also been evaluated [18].

It is also known that genetic variants can be a risk or protective factor against different diseases or infections and are associated with different degrees of susceptibility. Reports have suggested that environmental factors and genetic mutations play an essential role in the development of SARS-CoV-2 infection [19].

However, genetic mutations in *NFE2L2-KEAP1* genes responsible for activating the antioxidant and anti-inflammatory response could affect the signaling pathway, causing phenotypes with greater susceptibility to infection and even more severe COVID-19 [20]. Our results constitute one of the first pieces of evidence of the relationship of *NFE2L2* (rs2364723C/G) and *KEAP1* (rs9676881A/G; rs34197572C/T) with COVID-19.

In the case of rs2364723C/G, we found that the GG genotype had a higher genotypic and allelic frequency in control patients compared with patients with severe infection, suggesting that the minor allele (G) confers protection against SARS-CoV-2 infection. Different in vivo and in vitro studies have suggested that COVID-19 infection alters the redox balance within the cell and induces ROS production, causing the expression of pro inflammatory cytokines and the innate immune response. To mitigate the damage caused by oxidative stress, the NFR2-KEAP1 pathway is activated [21]. A codominance analysis was carried out to obtain greater statistical power. We found that the recessive model for the SNP *NFE2L2* (rs2364723) presents a greater risk when the GG genotype is diminished or absent, thus confirming that the minor allele confers protection against SARS-CoV-2 infection. There are no previous reports associating this polymorphism with COVID-19; however, *NFE2L2* polymorphisms may modulate the expression of the transcription factor or affect its ability to translocate into the nucleus and bind to the ARE site in target gene promoters [22]. The rs2364723C/G is in an intron and may cause changes in mRNA splicing and alterations in protein isoforms, which could modify its interaction with KEAP1, resulting in a significant alteration in the NFR2-KEAP1 signaling pathways, causing genotypes with greater susceptibility to viral infection.

Although little information exists on this SNP and its association with COVID-19, several studies have demonstrated the involvement of *NFE2L2* polymorphisms, which encode the NRF2 transcription factor, in cases of atherosclerosis, bronchial asthma, and chronic obstructive pulmonary disease [23,24]. Interestingly, SNP rs2364723 has been associated with a lower lung function level (FEV1), consistent with a severe COVID-19 phenotype [24].

Reactive oxygen species (ROS) are present in the pathogenesis of several lung diseases; in COVID-19, hypoxia is one of the primary triggers of pulmonary severity. In mice, strain-specific variation in lung *NFE2L2* mRNA expression and a T>C substitution in the *NFE2L2* promoter segregated with susceptibility phenotypes in F2 animals support the idea that NRF2 has important implications in understanding the mechanisms by which oxidants mediate the pathogenesis of lung disease [25,26,27].

*KEAP1* polymorphisms, rs9676881A/G and rs34197572C/T, were also associated with SARS-CoV-2 infection. In the first of these, we found a higher genotypic and allelic frequency (GG) in patients with moderate COVID-19 compared to the control group, a finding which might indicate that the G allele is a risk factor (OR 2.27) for developing moderate COVID-19. Few studies have been carried out on this polymorphism; however, it is known that rs9676881 is located 16 bp downstream in the 3′-untranslated region (UTR) of the *KEAP1* gene, near to transcription factor binding and enhancer part [28].

In the case of the SNP rs34197572C/T, we identified an association as follows: in the presence of the T allele, there is a high risk of severe COVID-19. In this SNP, the frequency of the minor allele in our study population was low, and although the results revealed significant differences, no forcefulness was demonstrated. In addition, the SNP rs34197572C/T is in the region 3′-UTR [29].

The possible mechanisms involved in those polymorphisms that cause a greater risk of infection include (1) mutations in the 3′UTR region, which affect the interaction with transcriptional regulatory molecules such as micro RNAs (miRs), causing deregulation in protein levels; (2) mutations near to the promoter sequence of a gene, which cause changes in the affinity of Pol II for the promoter site, generating a decrease in mRNA. To corroborate these ideas, it is necessary to quantify protein and mRNA levels. When reviewing the genotypic analysis models, we found that in both *KEAP1* SNPs, the recessive model confirmed that the minor allele is a risk factor for developing SARS-CoV-2 infection.

The KEAP1-NRF2-ARE pathway is one of the most critical mechanisms against oxidative stress; it is involved in chronic obstructive pulmonary disease and COVID-19 infection [30]. Some reports have suggested that mutations in KEAP1 are involved in the production of antioxidants, which could answer why some patients have more severe COVID-19 infections. It is important to note that the frequencies of the polymorphisms studied vary between populations. According to the HapMap genome, for rs9676881, where there is a substitution of A/G, the frequency globally is 0.49, while in the European population it is 0.36. However, in our study, the frequency was 0.77 in the controls and 0.82 in the COVID-19 patients. This indicates that the genetic load in the different populations differs markedly. In summary, we analyzed the associations of the NRF2/KEAP1 signaling pathway with SARS-CoV-2 infection in a population from Mexico. The data obtained indicate the contribution of the *NFE2L2-KEAP1* polymorphisms in patients with moderate or severe COVID-19 pneumonia. The research aimed to understand the Mexican hereditary predisposition to SARS-CoV-2 infection and thus help to identify new therapeutic targets against oxidative stress whose presence has been confirmed in this condition.

## 4. Materials and Methods

### 4.1. Research Population

A prospective study was carried out on a cohort of patients with COVID-19 between January 2020 and January 2021. A total of 111 patients with COVID-19 were recruited in the Critical Care Unit of the Temporal COVID-19 Unit, Citibanamex. The study population included patients who did or did not develop septic shock or experience moderate or severe pneumonia due to COVID-19. Diagnosis of septic shock was based on the Sepsis-3 consensus [31]. Patients considered to have experienced septic shock had to have at least a 2-point increase in SOFA score [32].

To verify infection by SARS-CoV-2 in the study design, samples from patients and personnel engaged in clinical and laboratory activities were collected. Paired saliva and nasopharyngeal swab samples were collected from 110 patients who were suspected to be infected by SARS-CoV-2. Samples were classified as positive for SARS-CoV-2 when the N1 and N2 primer-probe sets were detected. The presence of the SARS-CoV-2 virus was confirmed by the use of specific probes to detect the virus, in conjunction with the real-time reverse transcriptase polymerase chain reaction technique (qRT-PCR). To evaluate organ dysfunction, the SOFA score (neurologic, respiratory, hemodynamic, hepatic, and hematologic) was calculated at admission and during all treatment days.

The exclusion criteria in the COVID-19 group were as follows: patients under 18 years of age, those unable or unwilling to give informed consent, pregnant or lactating women, and those with a chronic (previous six months) or recent use of steroids, statins, or antioxidants.

In addition, 111 control patients were analyzed. The inclusion criteria for the control patients were that they had not suffered from COVID-19 prior to the time of the sample collection and that they did not suffer from hypertension or diabetes. Patients were recruited from the National Institute of Cardiology “Ignacio Chavez” blood bank. The exclusion criteria specified the presence of respiratory infections or of sexually transmitted infections.

### 4.2. Laboratory Analysis

Glucose, TC, and triglycerides were analyzed by enzymatic colorimetric methods (Roche-Syntex/Boheringer Mannhein, Mannheim, Germany). HDL-C was measured after precipitation of low- and very-low-density lipoproteins by phosphotungstate/Mg^2+^ (Roche-Syntex), and LDL-C was estimated by the equation of Friedewald et al. [33], modified by De Long et al. [34]. All the assays were under an external quality control scheme (Lipid Standardization Program, Center for Disease Control in Atlanta, GA, USA).

### 4.3. Anthropometric Measurement

Body mass index (BMI) was calculated as weight in kilograms divided by the square of height in meters (kg/m^2^). Patients were identified as overweight when BMI values were 25–29.9 and obese when BMI was ≥30. Plasma glucose was measured on an empty stomach; when the value was greater than 125 mg/dL, it was considered as indicating type 2 DM or a previous diagnosis with type 2 DM. Blood pressure was measured using a mercury sphygmomanometer following the recommendations of the VII Joint National Committee on Prevention, Detection, Evaluation, and Treatment of High Blood Pressure (JNC VII). Those in the hypertensive group had a blood pressure ≥ 140/90 mm Hg or had previously been diagnosed with essential hypertension. Dyslipidemia was indicated by increase in the plasma concentrations of cholesterol, triglycerides, or both, or by a decrease in the level of cholesterol associated with HDL (high-density lipoproteins).

### 4.4. COVID-19 Classification

The hospitalized patients who formed the general group were classified as moderate or severe, and this classification was based on their ventilatory status. According to the Berlin criteria for acute respiratory distress syndrome (ARDS), patients with severe conditions require invasive mechanical intubation.

The Berlin definition sets out three categories of ARDS based on the severity of hypoxemia, as follows: mild (200 mmHg < PaO_2_/FiO_2_ 300 mmHg), moderate (100 mm Hg < PaO_2_/FiO_2_ 200 mmHg), and severe (PaO_2_/FiO_2_ 100 mmHg) [35].

### 4.5. Ethical Approval

Ethical approval was obtained from the local ethics committee on 19 August 2020 (Control-9867/2020, register REG. CONBIOETICA-09-CEI-011-20160627). Written informed consent for enrollment or permission to use patient data was obtained from each patient or their legal surrogate. The protocol was registered (TRIAL REGISTRATION: ClinicalTrials.gov (accessed on 19 August 2020) Identifier: NCT04570254).

### 4.6. DNA Extraction

Genomic DNA was isolated from the blood samples using a commercial kit (Invitrogen Co., Carlsband, CA, USA). The DNA was quantified in a spectrophotometer (BioPhotometer plus) at a 260/280 nm wavelength.

### 4.7. Determination of Polymorphisms

The polymorphisms (Table 4) were determined using TaqMan probes in the CFX96^TM^ Touch Real-Time PCR Detection System. When allele-specific fluorogenic probes hybridize to the template during PCR, the activity of Taq polymerase can discriminate alleles. Cleavage results in increased emission of a reporter dye that is otherwise quenched by another dye. Each assay requires two unlabeled PCR primers and two allele-specific probes. Each probe is labeled with a reporter dye (VIC and FAM). The primers to detect *NFE2L2* (rs2364723) and *KAEP1* (rs9676881 and rs34197572) SNPs were synthesized by Applied Biosystems. The probe codes were C_35187810 for rs2364723; C__9323015_10 for rs9675881; and C_34043043_10 for rs34197572. TaqMan real-time PCR of *NFE2L2* (rs2364723), *KAEP1* (rs9676881), and *KAEP1* (rs34197572) polymorphisms was performed on an ABI Prism 7000 Sequence Detection System according to the manufacturer’s instructions (Applied Biosystems, FosterCity, CA, USA). A total of 6 μL of TaqMan^TM^ Universal PCR Master Mix was used in a reaction volume of 10 μL, at a final concentration of 10 ng of DNA, in addition to 700 nM of primers, and 100 nM of the probe labeled with fluorophores. The reaction conditions were 10 min at 95 °C and 40 cycles at 92 °C for 15 s, and 1 min at 60 °C. Each single nucleotide polymorphism (SNP) genotype and allele discrimination was both manually and automatically studied with the allelic discrimination software (7300 System SDS Software, Applied Biosystems).

### 4.8. Statistical Analysis

The Hardy–Weinberg equilibrium (HWE) for controls and patients was determined by means of a chi-square test. Polymorphism analysis was calculated using SPSS version 18 (SPSS Chicago, IL, USA) and the EPISTAT statistical program (Version 5.0; USD Incorporated 1990, Stone Mountain, Georgia). The *p* values were obtained according to the number of comparisons performed, and it was considered statistically significant if the *p*-value was <0.05. The relative risk was calculated as the odds ratio using 95% confidence intervals (CIs). Using codominance models, we evaluated the association of the SNPs studied and the severity of COVID-19 infection.

## 5. Conclusions

In conclusion, an association of NRF2/KEAP1 SNPs with COVID-19 pneumonia was identified. SNP NRF2 rs2364723C>G allele G demonstrated a protective effect against severe COVID-19, while KEAP1 SNP rs9676881A>G allele G and rs34197572C>T minor allele T were associated with more aggressive stages of COVID-19.

## Figures and Tables

**Table 1 ijms-24-00415-t001:** Anthropometric parameters of the study population.

Variable	COVID + Total (*n* = 110)	COVID + Severe (*n* = 51)	COVID + Moderate (*n* = 59)	Controls (*n* = 111)	p1	p2	p3	p4
Age (years)	58.00 ± 12.86	61.82 ± 12.37	54.53 ± 12.40	37.80 ± 14.32	<0.001	0.003	<0.001	<0.001
Gender % M/W	(78) 71%/(32) 29%	(37) 73%/(14) 27%	(41) 70%/(18) 30%	(73) 66%/(38) 34%	0.490	0.880	0.517	0.770
BMI (kg/m^2^)	29.14 ± 4.11	28.98 ± 3.80	29.27 ± 4.40	27.80 ± 4.34	0.018	0.713	0.091	0.035
Obesity %	(45) 41.0%	(21) 41.2%	(24) 40.7%	(38) 34.5%	0.333	0.692	0.001	0.009
Diabetes %	(42) 38.2%	(24) 47.1%	(18) 30.5%	—	—	0.319	—	—
Hypertension %	(44) 40.0%	(23) 45.1%	(21) 35.6%	—	—	0.843	—	—
Dyslipidemia %	(52) 47.3%	(20) 39.2%	(32) 54.2%	—	—	0.030	—	—

BMI—body mass index. The data are expressed as mean ± SD (Student’s *t*-test) or percentage (chi2 test). p1: COVID+ (total) vs. controls; p2: COVID+ (severe) vs. COVID+ (moderate); p3: COVID+ (severe) vs. controls; p4: COVID+ (moderate) vs. controls.

**Table 2 ijms-24-00415-t002:** Genotypic and allelic frequencies of the studied polymorphisms.

Gene	Genotype	COVID + Total	COVID + Severe n = 51	COVID + Moderate n = 59	Controls n = 111	p1	p2	p3	p4
rs2364723 *NFE2L2*	CC	48 (44%)	24 (47%)	24 (41%)	41 (37%)	0.37 1.32 (0.77–2.26)	0.63 1.29 (0.60–2.76)	0.29 1.51 (0.77–2.97)	0.75 1.17 (0.61–2.23)
	CG	46 (42%)	25 (49%)	21 (35%)	46 (41%)	0.93 1.01 (0.59–1.73)	0.21 1.73 (0.80–3.73)	0.46 1.35 (0.69–2.64)	0.56 0.78 (0.40–1.5)
	GG	16 (14%)	2 (4%)	14 (24%)	24 (22%)	0.23 0.61 (0.30–1.23)	0.001 0.13 (0.02–0.60)	0.001 0.14 (0.03–0.65)	0.90 1.12 (0.53–2.38)
Alleles	C	142 (65%)	73 (72%)	69 (58%)	128 (58%)	0.16 0.74 (0.50–1.09)	0.059 0.55 (0.31–0.98)	0.02 0.54 (0.32–0.89)	0.97 0.96 (0.61–1.52)
	G	78 (35%)	29 (28%)	49 (42%)	94 (42%)				
rs9676881 *KEAP1*	AA	4 (4%)	2 (5%)	2 (4%)	3 (2%)	0.99 1.35 (0.29–6.25)	0.71 1.16 (0.15–8.56)	0.94 1.46 (0.23–9.07)	0.97 1.53 (0.24–9.47)
	AG	32 (29%)	20 (39%)	12 (20%)	43 (39%)	0.16 0.64 (0.37–1.13)	0.04 2.52 (1.08–5.89)	0.90 1.02 (0.51–2.01)	0.02 0.40 (0.19–0.84)
	GG	74 (67%)	29 (56%)	45 (76%)	65 (59%)	0.22 1.45 (0.84–2.51)	0.04 0.41 (0.18–0.92)	0.97 0.93 (0.47–1.82)	0.03 2.27 (1.11–4.62)
Alleles	A	40 (18)	24 (24%)	16 (14%)	52 (23%)	0.24 1.35 (0.85–2.14)	0.08 0.50 (0.25–1.02)	0.95 0.97 (0.56–1.69)	0.04 1.91 (1.03–3.53)
	G	180 (82%)	78 (76%)	102 (86%)	173 (77%)				
rs34197572 *KEAP1*	CC	99 (90%)	41 (80%)	57 (97%)	106 (95%)	0.18 0.42 (0.14–1.26)	0.002 0.07 (0.00–0.57)	0.003 0.19 (0.06–0.60)	0.61 2.73 (0.31–23.9)
	CT	5 (5%)	4 (8%)	1 (1.5%)	2 (2%)	0.43 2.59 (0.49–13.6)	0.27 4.93 (0.53–45.6)	0.15 4.59 (0.81–25.96)	0.58 0.93 (0.08–10.4)
	TT	6 (5%)	6 (12%)	1 (1.5%)	3 (3%)	0.48 2.07 (0.50–8.52)	0.07 7.86 (0.91–67.6)	0.04 4.8 (1.14–20.03)	0.91 0.6 (0.06–5.99)
Alleles	C	203 (92%)	86 (84%)	117 (99%)	214 (96%)	0.09 2.24 (0.94–5.30)	0.004 21 (2.83–167)	0.001 4.97 (2.05–12.05)	0.13 0.22 (0.02–1.85)
	T	17 (8%)	16 (16%)	1 (1%)	8 (4%)				

p1: COVID + (Total) vs. Controls; p2: COVID + (several) vs. COVID + (moderate); p3: COVID + (several) vs. Controls; p4: COVID + (moderate) vs. Controls. OR: Odds Ratio.

**Table 3 ijms-24-00415-t003:** Overdominant, dominant, and recessive models of NEFL2 and KEAP1 SNP in COVID-19 patients.

Gene	Condition	Overdominant Model		Dominant Model		Recessive Model	
		OR (95% CI)	*p*	OR (95% CI)	*p*	OR (95% CI)	*p*
*NFE2L2* rs2364723	Severe vs. control	0.73 (0.37–1.43)	0.46	1.51 (0.77–2.97)	0.29	6.75 (1.53- 29.82)	0.008
	Moderate vs. control	1.28 (0.66–2.46)	0.56	1.17 (0.61–2.23)	0.75	0.88 (0.41–1.87)	0.90
*KEAP1* rs9676881	Severe vs. control	0.98 (0.49–1.93)	0.90	1.46 (0.23–9.07)	0.94	1.07 (0.54–2.09)	0.97
	Moderate vs. control	2.47 (1.18–5.19)	0.02	1.26 (0.20–7.77)	0.82	2.27 (1.11–4.62)	0.03
*KEAP1* rs34197572	Severe vs. control	0.21 (0.03–1.21)	0.14	5.17 (1.66–16.04)	0.005	4.8 (1.14–20.03)	0.04
	Moderate vs. control	1.08 (0.09–12.18)	0.58	1.36 (0.25–7.27)	0.97	0.62 (0.06–6.10)	0.90

OR: Odds Ratio. The OR, 95% CI, and *p* values of the genotypes were obtained from the codominant model.

**Table 4 ijms-24-00415-t004:** Information of the SNPs studied.

Gene SNP	*NFE2L2* rs2364723	*KEAP1* rs9676881	*KEAP1* rs34197572
Chromosome	2	19	19
SNP Type	Intron variant	Downstream variant	Downstream variant
Position	C107G	3′ UTR A>G	3′ UTR C>T
MAF	0.428	0.298	0.035

SNPs—single nucleotide polymorphisms. MAF—minor allele frequency. Data obtained from the 1000 Genomes Project.

## Data Availability

Not applicable.

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
