# Peer review of "Impact on the Clinical Evolution of Patients with COVID-19 Pneumonia and the Participation of the NFE2L2/KEAP1 Polymorphisms in Regulating SARS-CoV-2 Infection"

_ijms, 2022, doi:10.3390/ijms24010415_

Round 1
Reviewer 1 Report
The authors explored specific SNPs for their linkage with severe and moderate COVD19 patients compared to control uninfected Mexican population. Analysis looks interesting, however, some major improvements are required in manuscript as below:
1. Abstract need quite a bit of reformatting, for example as follows:
Line 33: “SNPs NFE2L2, rs2364723C>G, and KEAP1, rs9676881A>G and rs34197572C>T were determined by qPCR.” should be written as “SNPs in the gene locus for NFE2L2 (rs2364723C>G) and KEAP1 (rs9676881A>G and rs34197572C>T) were determined by qPCR.” Otherwise, sentence makes less sense.
Line 35-39 need to summarized.
Abstract should end with summary/conclusion/discussion line. It presently looks like just a mini-result section.
2. Introduction needs quite a bit of work as well. Nowhere it is mentioned where these SNPs fall in genome with respect to gene’s position until one reaches Discussion or Methods section. Simply writing, within intron or 3'UTR could have sufficed.
Moreover, nowhere in the introduction it is clearly mentioned that the NRF2 is also known as NFE2 like bZIP transcription factor 2 (NFE2L2), which is very crucial information for Introduction!
3. Why specifically those 3 SNPs chosen? Aren’t there any other SNPs in those gene’s locus, please explain it in the manuscript. What other manuscripts mentions those SNPs in lung disease context or Covid context.
4. What is full form of SOFA, please mention it somewhere in the Introduction.
5. Lines 120-128 at some places just mentions" significant difference found", but meaning of those differences could be made more clearer by mentioning whether a genotype is over-represented or under-represented in Severe covid patients.
6. For rs9676881 KEAP1 SNP, how can you explain why ‘GG’ or ‘AA’ genotype have same frequencies for both severe and control group. It seems the AA’ genotype is highly rare in general, it is found in only 4% population. So, the main comparison left is between ‘AG’ and ‘GG’ only. Data shows in that case having ‘GG’ is beneficial as it makes Covid more moderate, while having ‘AG’ leads to severe covid outcome. This concept doesn’t come out as clear as it should be in the manuscript. In this case minor ‘Allele’ is A and is associated with more aggressive stage (as written in Abstract last line), but the explanation is lacking. Moreover, the text is using “rs9676881A>G”, shouldn’t it be “rs9676881G>A” instead, please explain?
7. Line 177, please mention the SNP name (rs...) as well along with genotype.
8. Line 182, another explanation could be enhancers. Almost half of the enhancers are found in the intronic regions. Please check if this region with SNP has any H3K27ac mark in any cell type, especially lung cells.
9. Line 191-192, it is very small paragraph. Either join it somewhere else or expand on it (so that a new paragraph on its own could be justified).
10. Line 194 says “in the B6 NFE2L2 promoter segregated with susceptibility phenotypes”. What does “B6” means?
11. Line 195: “supported NRF2 has important implications understanding the mechanisms through which oxidants mediate the pathogenesis of lung disease”. Add “in” in the sentence as “important implications in understanding the”.
Author Response
Thank you to the reviewers for their comments:
Changes in the manuscript are marked in yellow
The manuscript has been sent and received by the language editing and review service, by MDPI English editing.
Reviewer 1.
The authors explored specific SNPs for their linkage with severe and moderate COVD19 patients compared to control uninfected Mexican population. Analysis looks interesting, however, some major improvements are required in manuscript as below:
- Abstract need quite a bit of reformatting, for example as follows:
ANSWER: This point has been corrected and reformatting in the text
Line 33: “SNPs NFE2L2, rs2364723C>G, and KEAP1, rs9676881A>G and rs34197572C>T were determined by qPCR.” should be written as “SNPs in the gene locus for NFE2L2 (rs2364723C>G) and KEAP1 (rs9676881A>G and rs34197572C>T) were determined by qPCR.” Otherwise, sentence makes less sense.
ANSWER: This point has been corrected in the text as suggested
Line 35-39 need to summarized.
ANSWER: This point has been corrected in the text
Abstract should end with summary/conclusion/discussion line. It presently looks like just a mini-result section.
ANSWER: This point has been corrected in the abstract
In conclusion, our results showed that NFE2L2 rs2364723C>G allele G has a protective effect against severe COVID-19, while KEAP1 rs9676881A>G, rs34197572C>T minor allele was associated with more aggressive stages of COVID-19.
- Introduction needs quite a bit of work as well. Nowhere it is mentioned where these SNPs fall in genome with respect to gene’s position until one reaches Discussion or Methods section. Simply writing, within intron or 3'UTR could have sufficed.
ANSWER: This point has been added in the text
so the purpose of this study was describe the participation of the single nucleotide polymorphisms (SNPs) of the NFE2L2 (rs2364723) localized in the intronic region and KEP1 (rs967688 and rs34197572) genes localized in the 3`UTR region, in lung disease in patients with SARS-CoV-2 infection.
Moreover, nowhere in the introduction it is clearly mentioned that the NRF2 is also known as NFE2 like bZIP transcription factor 2 (NFE2L2), which is very crucial information for Introduction!
ANSWER: This point has been added in the text
The nuclear factor erythroid 2- related factor 2 (known as NRF2 or NFE2). It comprises a pivotal system of cytoprotective responses to oxidative stresses, in which NFE2L2 is a master transcription factor that controls antioxidant enzymes [8]. Additionally, NFE2L2 is negatively regulated by the cysteine-rich protein KEAP1 via proteasomal degradation mediated by the CUL3 (cullin 3)-E3 ubiquitin ligase RBX1 (ring-box 1) complex. [9,10]. This process can result in the oxidation of cysteine residues and other proteins involved in viral infection.
- Why specifically those 3 SNPs chosen? Aren’t there any other SNPs in those gene’s locus, please explain it in the manuscript. What other manuscripts mentions those SNPs in lung disease context or Covid context.
These SNPs were chosen because they have been related to oxidative stress, and several publications have tried to elucidate their participation. We performed other SNPs in addition to those reported here, however, there were no changes in the frequencies, since all were homozygous. So far there are no reports of these polymorphisms with Covid-19.
This reference is one of the few in this regard with lung infection:
Functional polymorphisms in the transcription factor NRF2 in humans increase the risk of acute lung injury.
Marzec JM, Christie JD, Reddy SP, Jedlicka AE, Vuong H, Lanken PN, Aplenc R, Yamamoto T, Yamamoto M, Cho HY, Kleeberger SR. FASEB J. 2007 Jul;21(9):2237-46. doi: 10.1096/fj.06-7759com. Epub 2007 Mar 23.PMID: 17384144
- What is full form of SOFA, please mention it somewhere in the Introduction.
ANSWER: This point has been added in the text
to evaluate organ dysfunction the sequential organ failure assessment (SOFA) scoring was developed to provide a means of quantitatively and objectively describing the degree of organ failure over time in individual patients with sepsis. Therefore, SOFA score is calculated at admission and during all days of patient treatment.
- Lines 120-128 at some places just mentions" significant difference found", but meaning of those differences could be made more clearer by mentioning whether a genotype is over-represented or under-represented in Severe covid patients.
ANSWER: This point has been corrected and added
patients which presented an underrepresentation (80%) versus moderate group (97%) (OR 0.07, CI 95% (0.00-0.57, p= 0.002) and in control group (95%) (OR 0.19 CI 95% (0.06-0.60, p=0.003).
- For rs9676881 KEAP1 SNP, how can you explain why ‘GG’ or ‘AA’ genotype have same frequencies for both severe and control group. It seems the AA’ genotype is highly rare in general, it is found in only 4% population. So, the main comparison left is between ‘AG’ and ‘GG’ only. Data shows in that case having ‘GG’ is beneficial as it makes Covid more moderate, while having ‘AG’ leads to severe covid outcome. This concept doesn’t come out as clear as it should be in the manuscript. In this case minor ‘Allele’ is A and is associated with more aggressive stage (as written in Abstract last line), but the explanation is lacking. Moreover, the text is using “rs9676881A>G”, shouldn’t it be “rs9676881G>A” instead, please explain?
ANSWER: Thanks for the observation. Your comment regarding the frequencies for both groups is correct. The reason why the frequencies coincide, we don't know; It could be due to the genetic load between the populations, however we do not have a clear explanation. However, these frequencies vary according to the study population, in our population there are few studies in this regard. On the other hand, according to HapMap, the frequencies reported for the A allele worldwide is 0.51, while for the American population it is 0.30. According to HapMap, the change is from: GGTTTCAG[A/G]CCTC. However, your comment for our population could be the opposite as you mention.
- Line 177, please mention the SNP name (rs...) as well along with genotype.
ANSWER: This point has been corrected and added
- Line 182, another explanation could be enhancers. Almost half of the enhancers are found in the intronic regions. Please check if this region with SNP has any H3K27ac mark in any cell type, especially lung cells.
ANSWER: H3K27ac is an enhancer that has been reported in various cell types such as bone, hepatocytes or cancer cells, however, to date we have not found evidence of this enhancer in lung cells or those related to Covid-19
- Line 191-192, it is very small paragraph. Either join it somewhere else or expand on it (so that a new paragraph on its own could be justified).
ANSWER: This point has been corrected
- Line 194 says “in the B6 NFE2L2 promoter segregated with susceptibility phenotypes”. What does “B6” means?
ANSWER: This was a mistake, it's been removed
- Line 195: “supported NRF2 has important implications understanding the mechanisms through which oxidants mediate the pathogenesis of lung disease”. Add “in” in the sentence as “important implications in understanding the”.
ANSWER: This point has been added

Reviewer 2 Report
It is very important to detect and associate SNPs with illness or symptoms of a disease, let alone COVID-19. In this context, the study subject is important. However the manuscript is problematic taking into account the English language and more importantly the methodological framework.
Particularly, the written English is not appropriate in some parts. Several parts of the manuscript are very hard to follow. For instance the first sentence in the introduction starts at line 47 and ends at line 52. It has to be rephrased and separated in simpler phrases. Similarly, lines 230-231 need rephrasing, there is no verb, as well lines 231-232.
Methodological approach
In lines 239-242 there are no details for the qRT-PCR and the methodology applied for detection of primers-probes. The way it is written is not correct.
In lines 270-271 there are no details for the saline expulsion technique not even at least a reference for the method. How was it applied?
Similarly, in lines 276-277 the authors refer to the probes used, however the primers and probes sequences are not provided, nor a relevant reference is mentioned.
In lines 280-281 the authors state that the PCR products were quantified using… However the scope of the study was not quantification but to describe the SNPs. Also, if quantification has to be done there should be used a standard curve based on a control of known quantity or at least a created one. There is no such thing here
In the results, the authors state the body mass index, obesity etc, but there is no such description of measurement in the Materials and Methods
Finally I cannot see any evidence for overdominance.
Based on the above methodological limitations, I am afraid I cannot recommend publication of the study.
Author Response
Thank you to the reviewers for their comments:
Changes in the manuscript are marked in yellow
Reviewer 2.
It is very important to detect and associate SNPs with illness or symptoms of a disease, let alone COVID-19. In this context, the study subject is important. However the manuscript is problematic taking into account the English language and more importantly the methodological framework.
Particularly, the written English is not appropriate in some parts. Several parts of the manuscript are very hard to follow. For instance the first sentence in the introduction starts at line 47 and ends at line 52. It has to be rephrased and separated in simpler phrases. Similarly, lines 230-231 need rephrasing, there is no verb, as well lines 231-232.
ANSWER: The manuscript has been sent and received by the language editing and review service, by MDPI English editing.
Methodological approach
In lines 239-242 there are no details for the qRT-PCR and the methodology applied for detection of primers-probes. The way it is written is not correct.
ANSWER: This point has been corrected and added.
In lines 270-271 there are no details for the saline expulsion technique not even at least a reference for the method. How was it applied?
ANSWER: This point has been corrected:
Genomic DNA was isolated from the blood samples using a commercial kit (Invitrogen Co., Carlsband, CA, USA). The DNA was quantified in a spectrophotometer (BioPhotometer plus) at a 260/280 nm wavelength.
Similarly, in lines 276-277 the authors refer to the probes used, however the primers and probes sequences are not provided, nor a relevant reference is mentioned.
ANSWER: This point has been corrected:
When allele-specific fluorogenic probes hybridize to the template during PCR, the activity of Taq polymerase can discriminate alleles. Cleavage results in increased emission of a reporter dye that is otherwise quenched by another dye. Each assay requires two unlabeled PCR primers and two allele-specific probes. Each probe is labeled with a reporter dye (VIC and FAM). The primer to detect NFE2L2 (rs2364723) and KAEP1 (rs9676881 and rs34197572) SNPs were synthesized by Applied Biosystems. The probles code were C_35187810 for rs2364723; C__9323015_10 for rs9675881; C_34043043_10 for rs34197572. TaqManR real time PCR of the NFE2L2 (rs: 2364723), KAEP1 (rs: 9676881), and KAEP1 (rs:34197572) polymorphisms was performed on an ABI Prism 7000 Sequence Detection System according to manufacturer’s instructions (Applied Biosystems, FosterCity, CA, USA).
In lines 280-281 the authors state that the PCR products were quantified using… However the scope of the study was not quantification but to describe the SNPs. Also, if quantification has to be done there should be used a standard curve based on a control of known quantity or at least a created one. There is no such thing here
ANSWER: This point has been corrected.
In the results, the authors state the body mass index, obesity etc, but there is no such description of measurement in the Materials and Methods
ANSWER: This point has been added
4.2 Laboratory analysis
Glucose, TC and triglycerides were analyzed by enzymaticcolorimetric methods (Roche-Syntex/Boheringer Mannhein, Mannheim, Germany). HDL-C was measured after precipitation of low- and very-low-density lipoproteins by phosphotungstate/Mg2+ (Roche-Syntex) and LDL-C was estimated by the equation of Friedewald et al. [16], modified by De Long et al. [17]. All assays were under an external quality control scheme (Lipid Standardization Program, Center for Disease Control in Atlanta, GA, USA).
4.3 Anthropometric measurement
Body mass index (BMI) as calculated as weight in kilograms divided by the square height in meters (kg/m2), The overweight was calculated when BMI >25 to 29.9 and obesity when BMI ≥30. Plasma glucose is measured on an empty stomach; when the value was greater than 125 mg/dL it was considered as type 2 DM or have previously been diagnosed with type 2 DM. Blood pressure was measured using a mercury sphygmomanometer following the recommendations of the VII Joint National Committee on Prevention, Detection, Evaluation and Treatment of High Blood Pressure (JNC VII). For the hypertensive group, have a blood pressure ≥ 140/90 mmHg or have previously been diagnosed with essential hypertension. Dyslipidemia was considered when an increase in plasma concentrations of cholesterol, triglycerides or both was reported, or a decrease in the level of cholesterol associated with HDL (high-density lipoproteins)
Round 2
Reviewer 2 Report
I am not fully satisfied with all answers of the authors. Particularly, although numerous modifications were performed in the manuscript, how did the authors correct the part 280-281 "PCR products were quantified"? Did they completely delete it? The response to this comment is only "This point has been corrected." but I can not find how and where.
Also there is response to my comment regarding overdominance. The authors should explain it more, why they propose that it was not dominance?
Author Response
Thank you to the reviewers for their comments:
Reviewer 2.
I am not fully satisfied with all answers of the authors. Particularly, although numerous modifications were performed in the manuscript, how did the authors correct the part 280-281 "PCR products were quantified"? Did they completely delete it? The response to this comment is only "This point has been corrected." but I can not find how and where.
Answer. sorry for not clarifying it correctly.
That paragraph was removed because it was an error on our part. PCR products were not quantified as we were not seeing expression as in mRNA for example. In this case, PCR was performed for allelic discrimination.
Allelic discrimination is the process by which two variants of a single nucleotide sequence are detected in a sample. SNPs (Single Nucleotide Polymorphisms) are variations at a certain point in the nucleotide sequence of two individuals.
We, as corrected in the text, use Taqman probe technology to study SNPs. In this case, the two Taqman probes present in the assay are each complementary to each of the SNPs. Each has a different fluorochrome at the 5' end and a quencher at the 3' end. During the extension phase of the PCR reaction, the DNA polymerase breaks the probe(s) hybridized with the DNA, separating the quencher's fluorochrome and detecting fluorescence emission from one or both of the probes.
Add in the text: 4.7. Determination of polymorphisms
The polymorphisms (Table 4) were determined using TaqMan probes in the CFX96TM Touch Real-Time PCR Detection System. When allele-specific fluorogenic probes hybridize to the template during PCR, the activity of Taq polymerase can discriminate alleles. Cleavage results in increased emission of a reporter dye that is otherwise quenched by another dye. Each assay requires two unlabeled PCR primers and two allele-specific probes. Each probe is labeled with a reporter dye (VIC and FAM). The primers to detect NFE2L2 (rs2364723) and KAEP1 (rs9676881 and rs34197572) SNPs were synthesized by Applied Biosystems. The probe codes were C_35187810 for rs2364723; C__9323015_10 for rs9675881; and C_34043043_10 for rs34197572. TaqMan real-time PCR of NFE2L2 (rs: 2364723), KAEP1 (rs: 9676881), and KAEP1 (rs:34197572) polymorphisms was performed on an ABI Prism 7000 Sequence Detection System according to the manufacturer’s instructions (Applied Biosystems, FosterCity, CA, USA). A total of 6 μL of TaqManTM Universal PCR Master Mix was used in a reaction volume of 10 μL, at a final concentration of 10 ng of DNA, in addition to 700 nM of primers, and 100 nM of the probe labeled with fluorophores. The reaction conditions were 10 min at 95 ° C and 40 cycles at 92 ° C for 15 s, and 1 min at 60 °C. Each single nucleotide polymorphism (SNP) genotype and allele discrimination was both manually and automatically studied with the allelic discrimination software (7300 System SDS Software, Applied Biosystems).
Also there is response to my comment regarding overdominance. The authors should explain it more, why they propose that it was not dominance?
Answer: Sorry, but I don't quite understand the question.
In genotypic analysis models, there are different models such as:
Dominant (which compares AA vs AB + BB)
Recessive (which compares BB vs AA + AB)
Codominant 1 (which compares AA vs AB)
Codominant 2 (which compares AA vs BB)
Additive (which compares 2(BB) + AB vs AA
Overdominant (which compares AA + BB vs BB)
Allelic (which compares allele A vs B)
Therefore, each one is a different analysis of the genotypes, and that can explain different associations.
In our study, the analysis was performed in the dominant, recessive and overdominant models between the 2 study groups (Table 3).
In which we found significant differences as mentioned in the text:
we found significant differences for moderate infection vs. control groups in both overdominant and recessive models ((OR 2.47 IC 95% (1.18-5.19), p=0.02) and (OR 0.43 IC 95% (0.21-0.89), p=0.03), respectively). Statistically significant differences were also found in the SNP KEAP1 (rs34197572) when comparing the severe COVID-19 group against the controls in the dominant and recessive models ((OR 0.19 IC 95% (0.06-0.60), p=0.005), and (OR 0.20 IC 95% (0.04-0.86), p=0.04), respectively) (Table 3).
Our results showed different associations depending on the model analyzed.
I don't know if this is what the reviewer is referring to?

Round 3
Reviewer 2 Report
The manuscript can be published in its current form